# Impact of Preprocedural Collateral Status on Hemorrhagic Transformation and Outcomes After Endovascular Thrombectomy in Acute Ischemic Stroke [note 1]

**DOI:** 10.3390/diagnostics15212701

**Published:** 2025-10-25

**Authors:** Shiu-Yuan Huang, Nien-Chen Liao, Jin-An Huang, Wen-Hsien Chen, Hung-Chieh Chen

**Affiliations:** 1Department of Radiology, Taichung Veterans General Hospital, Taichung 407219, Taiwan; stevenhuang19951203@gmail.com; 2Department of Neurology, Neurological Institute, Taichung Veterans General Hospital, Taichung 407219, Taiwan; rufus0822@gmail.com (N.-C.L.); jahuang@vghtc.gov.tw (J.-A.H.); 3Institute of Clinical Medicine, National Yang Ming Chiao Tung University, Taipei 112304, Taiwan; 4Department of Health Business Administration, Hungkuang University, Taichung 433304, Taiwan; 5Department of Post-Baccalaureate Medicine, College of Medicine, National Chung Hsing University, Taichung 402202, Taiwan; 6College of Medicine, National Yang Ming Chiao Tung University, Taipei 112304, Taiwan

**Keywords:** hemorrhagic transformation, endovascular thrombectomy, acute ischemic stroke, collateral status, recanalization, Alberta Stroke Program Early CT Score

## Abstract

**Background**: Hemorrhagic transformation (HT) is a major complication of endovascular thrombectomy (EVT) for acute ischemic stroke (AIS). **Objectives**: To investigate the factors as sociated with HT in patients with successful recanalization and examine the impact of collateral status (CS) on ischemic progression and outcomes. **Methods**: We retrospectively analyzed patients with AIS with successful recanalization (modified treatment in cerebral infarction (mTICI) 2B-3) who underwent dual-energy CT (DECT) within 24 h and MRI within 10 days post-EVT. Patients with posterior circulation stroke, missing multiphase CT angiography (CTA) collateral scores, or missing 3-month modified ranking scale scores were excluded from the study. **Results**: Among the 86 patients, those with HT had a significantly lower proportion of 3-month excellent outcomes and worse imaging scores, including non-contrast CT (NCCT)-Alberta Stroke Program Early CT Score (ASPECTS), virtual non-contrast (VNC)-ASPECTS, and diffusion-weighted imaging (DWI)-ASPECTS. Patients with HT with poor CS had a significantly lower proportion of 3-month excellent outcomes, poorer post-EVT National Institutes of Health Stroke Scale (NIHSS) score, worse imaging scores, including VNC-ASPECTS, and DWI-ASPECTS. In the predictive factor analysis, post-EVT NIHSS and VNC-ASPECTS scores were significantly associated with 3-month excellent functional outcomes (modified Rankin Scale (mRS) 0-1). **Conclusions**: In patients with successfully recanalized AIS, HT with poor CS was associated with poorer functional outcomes and worse imaging scores, and a 24 h combined measure (post-EVT NIHSS and DECT VNC-ASPECT) show promise for early risk stratification; prospective external validation is warranted before routine use.

## 1. Introduction

Endovascular thrombectomy (EVT) is the standard treatment option for selected patients with large vascular occlusion acute ischemic stroke (AIS) [1]. Hemorrhagic transformation (HT) is a common complication in patients with AIS managed with EVT, and specific subtypes, such as parenchymal hemorrhage, are associated with poor functional outcomes [2]. The mechanisms underlying HT are complex, involving ischemic reperfusion injury and blood–brain barrier disruption, and may be affected by the collateral status (CS) [3,4]. CS can be evaluated using different imaging tools, such as single-phase CT angiography (CTA), multiphase CTA, and MRI, and has been shown to help predict clinical outcomes [5]. Previous studies have found that a robust CS is associated with reduced infarct expansion and improved functional recovery [6,7,8]. A poor CS is associated with a higher risk of hemorrhagic complications and poor functional outcomes, with absent or poor collaterals related with hemorrhagic infarction, a subtype of hemorrhagic transformation [9,10,11]. In addition to HT and CS, both of which are associated with clinical outcomes, successful revascularization is an important predictor of good outcomes after EVT. Currently, the modified Thrombolysis in Cerebral Infarction (mTICI) score has been used to assess post-procedural recanalization according to cerebral perfusion [12,13,14].

The evaluation tools and the influence of CS vary among studies, with most focusing on patients with AIS, regardless of successful recanalization. In patients with successfully recanalized AIS managed with EVT, the impact of preprocedural CS is not well recognized. Hence, our study aimed to investigate the factors associated with HT in successfully recanalized patients and to examine the impact of CS on ischemic progression and outcomes. We hypothesized that poor preprocedural CS is associated with a higher risk of HT and worse outcomes in successfully recanalized patients with AIS.

## 2. Materials and Methods

### 2.1. Study Population

This retrospective cohort was conducted at Taichung Veterans General Hospital with institutional review board approval (CE24606A); the requirement for informed consent was waived owing to the study design. Consecutive patients treated between January 2019 and August 2024 were screened. Eligibility criteria were (1) acute ischemic stroke (AIS) treated with EVT; (2) the symptom-to-EVT interval less than 24 h; (3) pre-EVT National Institutes of Health Stroke Scale (NIHSS) score ≥ 6; (4) successful recanalization, defined as mTICI 2B-3; (5) dual-energy CT (DECT) within 24 h after EVT; and (6) follow-up MRI within 10 days after EVT. Patients were excluded for posterior circulation stroke, absence of a multiphase CTA collateral score, or missing 3-month follow-up modified Rankin Scale (mRS).

### 2.2. Imaging and Endovascular Treatment Protocols

All patients underwent non-contrast CT (NCCT), CTA, and CT perfusion (CTP) before EVT. EVT was performed by board-certified interventionalists using a biplane angiography system, following institutional protocols. Recanalization was graded on the mTICI scale; mTICI 2B-3 was considered successful [13]. An immediate cone-beam CT was acquired in the angiosuite to screen for large hemorrhage or procedure-related complications. Within 24 h of EVT, DECT was performed on a dual-source CT scanner (IQon Spectral CT, Philips Healthcare, Eindhoven, The Netherlands) to generate simulated 120 kV mixed images (sNCCT), virtual non-contrast (VNC) images, and iodine overlay maps (IOM) using a standard 3-material decomposition algorithm. DECT allows differentiation of hemorrhage from contrast staining and enables VNC/IOM-based quantification. Follow-up MRI was obtained within 10 days of EVT (median [IQR], 139 h) to establish the presence of HT. Because follow-up imaging was determined clinically, we restricted inclusion to patients with DECT within 24 h and follow-up MRI within 10 days after EVT procedure to reduce the time-point variability. For all acquisitions, patients were scanned supine with arms alongside their bodies.

### 2.3. Clinical Variables

Clinical information was abstracted from electronic medical records and the institutional imaging archives. Baseline variables collected before the procedure included demographics (age and sex), comorbidities, current medications, use of thrombolytics, NIHSS at the emergency department when arrival (ER NIHSS), pre-EVT laboratory tests, and occlusion site (left/right) and occlusion location (MCA, ICA, or combined ICA/MCA/ACA) determined by preoperative CTA. Periprocedural variables included use of percutaneous transluminal angioplasty or stenting, the mTICI grade, and cone-beam CT findings obtained immediately following EVT. Post-procedural variables included the 24 h NIHSS, any neurosurgical intervention, in-hospital mortality, length of stay during the EVT admission, functional outcomes accessed by the mRS, and 90-day mortality. The 3-month mRS values were obtained during outpatient visit or structured telephone interviews.

### 2.4. Imaging Analysis

Sequential imaging followed a uniform institutional pathway. Pre-EVT NCCT/CTA/CTP and post-EVT DECT/CT/MRI were reviewed by a postgraduate trainee and neuroradiologist with 16 years of experience. Both reviewers evaluated the images, and any discrepancies were resolved by consensus.

For quantitative analysis, the ischemic core volume (mL, CBF < 30%) and penumbral volume (mL, Tmax > 6 s) were extracted using RAPID v3.22.0 (iSchemaView, Golden, CO, USA) automated software [15]. The CS on multiphase CTA was graded on a six-point ordinal scale; good CS was defined as 4–5, and poor CS as 0–3 [5]. The Alberta Stroke Program Early CT Score (ASPECTS) was assigned on pre-EVT NCCT, and on post-EVT sNCCT, post-EVT VNC, and post-EVT DWI images yielding the NCCT-ASPECTS, sNCCT-ASPECTS, VNC-ASPECTS, and DWI-ASPECTS scores [16]. Hyperdense regions on the post-EVT IOM were also scored according to the ASPECTS (IOM-ASPECTS) [17]. The ischemic volume (mL) on follow-up MRI was measured on DWI/ADC. The final diagnosis of HT was based on the follow-up MRI.

### 2.5. Outcome Measures

Clinical outcomes comprised the need of neurosurgical intervention, in-hospital length of stay during the EVT admission, and 90-day functional outcome and mortality. Ninety-day outcomes were summarized as favorable (mRS 0–2) and excellent (0–1).

Imaging outcomes included ischemic region progression and ischemic volume growth. Progression was summarized as the differences between DWI-ASPECTS and earlier modality ASPECT: NCCT-ASPECTS (NCCT-DWI-ASPECTS), and VNC-ASPECTS (VNC-DWI-ASPECTS). Volume growth was defined as follow-up MRI ischemic volume—pre-EVT core volume (CTP < 30%).

### 2.6. Statistical Analysis

Continuous variables were summarized as medians (interquartile ranges, IQR) and subgroups were compared with the Mann–Whitney U test. Categorical variables were expressed as percentages and compared using the Chi-square test or Fisher’s exact test.

We first used univariate logistic regression analyses to evaluate associations with 90-day favorable and excellent outcomes. Variables with a *p*-value < 0.05 advanced to multivariable logistic regression. Two-sided significance was set at *p* < 0.05. For continuous parameters associated with functional outcomes, we conducted receiver operating characteristic (ROC) curves. Analyses were performed in SPSS software v.29 (IBM Corp., Armonk, NY, USA).

## 3. Results

### 3.1. Patient Characteristics and Clinical and Imaging Profiles

Between January 2019 and August 2024, 454 patients underwent EVT for acute stroke at our hospital. We excluded 190 patients who did not undergo DECT within 24 h, 35 patients with posterior circulation stroke, 99 patients who did not undergo follow-up MRI within 10 days after EVT, 20 patients without collateral score evaluation, 23 patients with mTICI 0-2A, and 1 patient without a 3-month follow-up; ultimately, 86 patients were included in our analysis (Figure 1). The cohort included 50 males (58.1%) and 36 females (41.9%), with a median age of 70 years (IQR 60–77 years). The median NIHSS score at admission was 16.5 (IQR 13–20), and the median NIHSS score observed 24 h after EVT was 10 (IQR 6–16). Regarding comorbidities, 52 (60.5%) patients had hypertension, 33 (38.4%) had diabetes mellitus, 63 (73.3%) had dyslipidemia, and 44 (51.2%) had atrial fibrillation. A total of 38 patients (44.2%) underwent IV thrombolysis before EVT. After EVT, 33 patients (38.4%) with no HT developed, and 53 patients (61.6%) developed HT. Among the patients who developed HT, 35 (40.7%) had good CS and 18 (20.9%) had poor CS (Table 1).

For the preprocedural image profile, the median CBF < 30% volume was 16 mL (IQR 0–43 mL), median Tmax > 6 s was 98 mL (IQR 68–131 mL), median collateral score was 4 (IQR 3–4), and median NCCT-ASPECTS was 9 (IQR, 8–9). For post-procedural DECT images, the median sNCCT-ASPECTS was 8 (IQR 8–9), the median VNC-ASPECTS was 6.5 (IQR 5–8), and the median IOM-ASPECTS was 6 (IQR 6–8). For the post-procedural MRI, the median DWI-ASPECTS was 6 (IQR 4–7), and the median ischemic volume was 16.6 mL (IQR 5.1–41.6 mL) (Table 1).

For the clinical outcome analysis, 5 patients (5.8%) underwent neurosurgical intervention, median hospital stay of 14.5 days (IQR 9–19 days), 41 patients (47.7%) had favorable 3-month functional outcomes, 26 patients (30.2%) had excellent 3-month functional outcomes, and 7 patients (8.1%) had died at 3 months. For image outcome analysis, the median image region progression was 3 (IQR 1–4) for the NCCT-DWI-ASPECTS and 0 (IQR 0–2) for the VNC-DWI-ASPECTS. The median ischemic volume growth was 3.3 mL (IQR −15.4 to 19.9 mL).

### 3.2. Hemorrhagic Transformation Development Analysis

To identify the factors correlated with HT in our patient cohort, we compared the clinical and imaging characteristics between the HT and non-HT groups. In the clinical parameter analysis, previous tPA (*p* = 0.016), cholesterol (*p* = 0.013), LDL (*p* = 0.012), post-24 h NIHSS (*p* = 0.005), and NIHSS improvement after EVT (*p* = 0.033) were significantly different between the HT and non-HT groups. Significant differences were observed for the in-hospital length of stay (*p* = 0.002) and the 3-month excellent mRS (*p* = 0.015). No differences were observed in terms of age, sex, comorbidity, reperfusion time, admission NIHSS, occlusion site, occlusion location, or the proportion of patients with favorable mRS (≤2) at 3 months between the two groups (Table 1).

For image parameter analysis, NCCT-ASPECTS (*p* = 0.005), sNCCT-ASPECTS (*p* < 0.001), VNC-ASPECTS (*p* < 0.001), and DWI-ASPECTS (*p* = 0.002) were significantly different between the HT and non-HT groups (Table 1).

### 3.3. Collateral Status and Hemorrhagic Transformation Analysis

We further divided the patients with HT into HT with good CS and HT with poor CS groups and compared them with the non-HT group. For clinical parameter analysis, previous tPA (*p* = 0.029), cholesterol (*p* = 0.004), PT (*p* = 0.024), APTT (*p* = 0.012) levels, onset-to-reperfusion duration (*p* = 0.02), and post-24 h NIHSS (*p* = 0.002) were significantly different between the HT with poor CS group, the HT with good CS group, and without HT group (Figure 2. Significant differences were observed regarding in-hospital length (*p* = 0.005) and 3-month excellent mRS (*p* = 0.029) (Figure 3). No significant difference was found between the occlusion site, occlusion location, NIHSS score at admission, and the 3-month favorable mRS (Table 2).

For image parameter analysis, the CBF < 30% volume (*p* = 0.004), NCCT-ASPECTS (*p* = 0.015), sNCCT-ASPECTS (*p* < 0.001), VNC-ASPECTS (*p* < 0.001), and DWI-ASPECTS (*p* < 0.001) were significantly different across the three groups (Figure 3). The ischemic region progression of the NCCT-DWI-ASPECTS (*p* = 0.003) was also statistically different (Table 2). Considering the potential influence of rescue angioplasty/stenting, we conducted a sensitivity analysis excluding all patients who underwent rescue PTA and/or stenting (final *n* = 68; Non-HT *n* = 28; HT *n* = 40) (Appendix A). The direction and magnitude of our main findings were preserved: (1) Post-EVT 24-h NIHSS remained higher in HT vs. non-HT (13.5 vs. 8, median; *p* = 0.012). (2) Post-EVT ASPECTS (sNCCT/VNC/DWI) remained lower in HT (all *p* ≤ 0.006).(3) Length of stay remained longer in HT (*p* < 0.001). (4) Excellent outcome (mRS ≤ 1) remained less frequent in HT vs. non-HT (10.0% vs. 42.9%, *p* = 0.002). Furthermore, across Non-HT, HT with good CS, and HT with poor CS, we observed graded worsening in 24-h NIHSS (8 vs 11 vs. 16.5, *p* = 0.006), post-EVT ASPECTS (all *p* ≤ 0.020; sNCCT/VNC/DWI *p* < 0.001), length of stay (*p* = 0.002), and excellent outcomes (42.9% vs. 12.5% vs. 6.3%, *p* = 0.006) (Appendix A).

### 3.4. Predictive Factors of Long-Term Favorable Functional Outcome: Univariate and Multivariate Analysis

In the univariate regression analysis, age (*p* = 0.007), NIHSS score at admission (*p* = 0.002), NIHSS score at 24 h post-EVT (*p* = 0.003), sNCCT-ASPECTS (*p* = 0.036), and VNC-ASPECTS (*p* = 0.012) were associated with favorable functional outcomes at 3 months after EVT. In the multivariate regression analysis, age (*p* = 0.011) was associated with favorable functional outcomes at 3 months after EVT (Table 3). We further performed ROC curve analysis for the post-NIHSS and VNC-ASPECTS. For the post-NIHSS, a cut-off value of 8 for identifying favorable outcomes resulted in an area under the curve of 0.710, with 68% sensitivity (95% CI, 52–82%), 77% specificity (95% CI, 61–88%), and a likelihood ratio of 2.94. For the VNC-ASPECTS, a cut-off value of 7 for identifying favorable outcomes showed an AUC of 0.659, with 63% sensitivity (95% CI, 47–78%), 62% specificity (95% CI, 47–76%), and a likelihood ratio of 1.68.

To explore whether a combination of clinical and imaging parameters could enhance prognostic accuracy, we constructed a combined score using both the post-EVT NIHSS and VNC-ASPECTS. Patients meeting both criteria (post-EVT NIHSS ≤ 8 and VNC-ASPECTS ≥ 7) were classified as having a positive combined score. For the combined score, the AUC was 0.724, with 56% sensitivity (95% CI, 40–72%), 84% specificity (95% CI, 69–93%), and a likelihood ratio of 3.45 (Figure 4).

### 3.5. Predictive Factors of Long-Term Excellent Functional Outcome: Univariate and Multivariate Analysis

In the univariate regression analysis, HT (*p* = 0.017), age (*p* = 0.04), NIHSS at admission (*p* = 0.006), NIHSS 24 h post-EVT (*p* = 0.001), NCCT-ASPECTS (*p* = 0.043), sNCCT-ASPECTS (*p* = 0.002), VNC-ASPECTS (*p =* 0.001), DWI-ASPECTS (*p* = 0.001), and ischemic region progression of NCCT-DWI-ASPECTS (*p* = 0.01) were associated with excellent functional outcomes at 3 months after EVT. In the multivariate regression analysis, post-NIHSS (*p* = 0.027) was associated with excellent functional outcomes at 3 months after EVT (Table 4). We further performed ROC curve analysis for the post-NIHSS and VNC-ASPECTS. For the post-NIHSS, a cut-off value of 9 for identifying excellent outcomes showed an area under the curve of 0.774, with 81% sensitivity (95% CI, 61–93%), 66% specificity (95% CI, 52–78%), and a likelihood ratio of 2.34. For the VNC-ASPECTS, a cut-off value of 7 was determined to identify excellent outcomes with an AUC of 0.753, 81% sensitivity (95% CI, 61–93%), 63% specificity (95% CI, 50–75%), and a likelihood ratio of 2.20. To evaluate whether combining clinical and imaging predictors improved discriminative performance, we defined a combined score that met both criteria as follows: post-EVT NIHSS score ≤ 9 and VNC-ASPECTS score ≥ 7. This combined score achieved an AUC of 0.813, with 88% sensitivity (95% CI, 70–98%), 65% specificity (95% CI, 52–78%), and a likelihood ratio of 2.57 (Figure 4).

## 4. Discussion

This study examined how CS and HT related to clinical outcomes and imaging profiles. We found that HT with poor CS was associated with worse functional outcomes and imaging scores, consistent with more severe ischemic injury and neurological deterioration. Furthermore, post-EVT NIHSS (24 h) and VNC-ASPECTS were associated with functional outcomes, highlighting their potential as imaging and clinical biomarkers for risk stratification.

HT is a frequent post-EVT complication and is linked to poor clinical outcomes [2,18]. In our cohort, HT following EVT was associated with substantially worse functional outcomes, particularly when preprocedural CS was poor. Van Kranendonk et al. reported that poor or absent collaterals were significantly associated with HT. In contrast, poor CS was not associated with HT in our dataset, a difference that may reflect limited sample size [9,10,11]. Patients with HT exhibited lower rates of excellent outcomes, higher 24 h post-EVT NIHSS, and worse image scores (NCCT/VNC/DWI-ASPECTS). Moreover, the HT with poor CS subgroup had the poorest outcomes and the lowest imaging scores compared to HT with good CS and non-HT groups. This pattern is consistent with the findings of Menon et al., who observed that poor-collateral flow accompanies greater ischemic injury and a higher risk of HT [5]. By jointly considering HT and CS, our analysis clarified their combined influence on long-term outcomes and imaging characteristics.

The 24 h post-EVT NIHSS score provides practical assessment of neurological function and relates to subsequent clinical outcomes. Our study found that the 24 h post-EVT NIHSS score was associated with both favorable and excellent 90-day functional outcomes; a threshold of 9 for identifying excellent outcomes showed an AUC of 0.774 (sensitivity 81%, specificity 66%). In our cohort, patients with HT and poor CS had the highest median post-EVT NIHSS scores, reflecting the combined impact of hemorrhage and an insufficient perfusion reserve on neurological function. Our finding aligns with previous study by Jeong et al., who suggested that the post-EVT NIHSS score can serve as an appropriate baseline factor when evaluating an intervention after a hyperacute period [19]. Saver et al. reported that serial NIHSS scores trajectories correlated with global disability and decreased over time [20]. Sajobi et al. found that the early trajectory of the NIHSS score within the first 48 h after EVT helps predict functional outcomes with high accuracy [21]. Together, these reports and our data support that utility of 24 h post-EVT NIHSS as a practical predictive clinical marker for long-term outcomes.

The ASPECTS remains a key topographic marker for estimating infarct burden and guiding treatment decision in patients with AIS [16]. In this cohort, all ASPECTS modalities, including NCCT-, VNC-, and DWI-ASPECTS, were lower in the HT group than in the non-HT group, with the nadir among patients with HT and poor collaterals. Among patients with successful recanalization, VNC-ASPECTS correlated with both hemorrhagic risk and functional outcome, consistent with our previous observations [22]. While pre-EVT NCCT-ASPECTS remains informative, our results underscore the complementary value of post-EVT VNC-ASPECTS for capturing changes linked to hemorrhagic risk and prognosis.

Prior work supports the clinical utility of pre-EVT NCCT-ASPECTS: Ahn et al. linked low pre-EVT NCCT-ASPECTS to intraparenchymal hemorrhage with poorer outcome [23]. Leker et al. reported the post-EVT ASPECTS ≥ 7 predicted good outcomes with high sensitivity [24]. Consistent with these reports, pre-EVT NCCT-ASPECTS in our cohort was associated with excellent functional outcomes.

VNC imaging enhances depiction of ischemic tissue in AIS after endovascular treatment and differentiates intracranial hemorrhage from contrast extravasation [25,26,27,28,29]. Van den Broek et al. showed that DECT angiography-derived VNC (DECTA-VNC) produced ASPECT scores comparable to NCCT with superior inter-rater agreement [30]. Our previous study also found that the post-EVT VNC-ASPECT was associated with both HT and favorable 3-month functional outcome [22]. For DWI-ASPECTS, previous studies mainly examined pre-treatment settings; Tei et al. showed that DWI-ASPECTS can be an independent predictor of functional outcome in patients with AIS, and Dogariu et al. found a strong negative correlation between DWI-ASPECTS and mRS scores [31,32]. Our results emphasize post-EVT reassessment: both VNC-ASPECTS and DWI-ASPECTS obtained after EVT were associated with 3-month functional outcomes. This reinforces its clinical utility and suggests that incorporating sequential pre-EVT NCCT-ASPECTS, post-EVT VNC-ASPECTS, and DWI-ASPECTS into routine EVT evaluation and treatment could enhance early prognosis and inform decisions regarding the need for critical care.

We quantified ASPECTS progression as the decline from baseline NCCT-ASPECTS to follow-up VNC and DWI. In our cohort, the median NCCT to DWI decline was three points, indicating notable infarct evolution despite successful reperfusion. Patients with poor-collateral HT showed the greatest ASPECTS declines across NCCT, VNC, and DWI, supporting the concept that collateral failure leads to accelerated tissue damage and impaired reperfusion recovery. These observations align with Xu et al., who related faster ASPECTS decay to worse 90-day outcomes and poor CS [33]. Tracking ASPECTS trajectories may aid post-EVT risk stratification.

However, ASPECTS offers regional topography, and infarct volume growth provides a global quantitative assessment of injury. In our study, the median infarct volume growth was 3.3 mL and did not differ significantly by the HT status or CS subgroups, nor did it correlate with 3-month functional outcomes. This pattern may reflect follow-up timing and sample size limitations and the possibility that volumetric measures miss regional changes captured by the ASPECTS.

Our findings are consistent with those of Ospel et al., who reported that infarcts continue to grow after EVT, even if near-complete reperfusion is achieved [34]. Robert et al. also found that infarct growth occurred despite adequate reperfusion, especially in the cortex, and independently decreased the odds of a good outcome [35]. A meta-analysis by Bala et al. showed higher infarct growth at higher baseline infarct volumes, with higher rates of incomplete reperfusion and longer onset-to-reperfusion times [36]. Liao et al. reported that a reduction in infarct growth was associated with an increased probability of functional independence [37]. However, our study showed that infarct growth volume was not associated with long-term outcomes, and that patients with poor collaterals and HT did not show significantly greater infarct growth. This result may be because of variations in the duration of follow-up MRI and fewer patient numbers compared with the studies mentioned. A conference abstract we previously presented based on the same institutional database but focused primarily favorable outcome as mRS 0–2 with collateral status, in our current study, we defines both favorable outcome (mRS 0–2) and excellent outcome (mRS 0–1), and further evaluates the predictive value of DECT VNC-ASPECTS (alone and in combination with post-EVT NIHSS), thereby addressing a different primary hypothesis and findings [38].

Based on our cohort results, a combination of imaging and clinical factors (post-NIHSS+VNC-ASPECTS) presented superior predictive ability for long-term functional outcomes when compared with imaging or clinical parameters alone. This suggests that a comprehensive pre- and post-EVT evaluation framework that integrates imaging and clinical factors could establish predictive models for long-term outcomes; hence, further prospective studies are warranted.

Our study had some limitations. First, because this was a retrospective study conducted at a single institution, there was an inherent risk of selection bias. Most of our patients were Taiwanese, limiting racial diversity and thereby restricting the generalizability of our findings. Second, although all patients underwent follow-up DECT within 24 h after endovascular thrombectomy (EVT), the timing of follow-up MRI varied depending on individual clinical conditions and physician judgment. This variability may have influenced the detection of HT. To reduce this bias, we excluded patients who underwent follow-up MRI > 10 days after EVT, but some cases of asymptomatic HT might still have been missed. Third, subgroup sizes—particularly HT with poor collaterals (*n* = 18)—were small, reducing statistical power for between-group comparisons and increasing uncertainty around effect estimates. Fourth, rescue angioplasty/stenting and periprocedural antithrombotic strategies may confound both HT and outcomes, and residual confounding cannot be excluded in an observational design. Finally, the combined criteria derived from 24 h post-EVT NIHSS and DECT-derived VNC-ASPECTS were developed and assessed in-sample; these thresholds may be optimistic and will require prospective, multicenter external validation—potentially across different scanners and post-processing platforms—before any routine implementation.

## 5. Conclusions

In conclusion, patients with AIS who achieved successful recanalization but experienced HT and poor CS had lower imaging scores (NCCT/VNC/DWI-ASPECTS) and poorer long-term functional outcomes. This suggests more extensive ischemic injury and greater neurological deterioration in this subgroup. In addition, a simple combination of 24 h post-EVT NIHSS and DECT-derived VNC-ASPECTS showed moderate in-sample discriminative ability for long-term outcomes (e.g., AUC ~0.81 for excellent outcome), suggesting potential utility for early post-EVT risk stratification beyond hemorrhage/contrast differentiation. Multicenter external validation is needed to verify thresholds and establish clinical utility across platforms and populations.

## Figures and Tables

**Figure 1 diagnostics-15-02701-f001:**
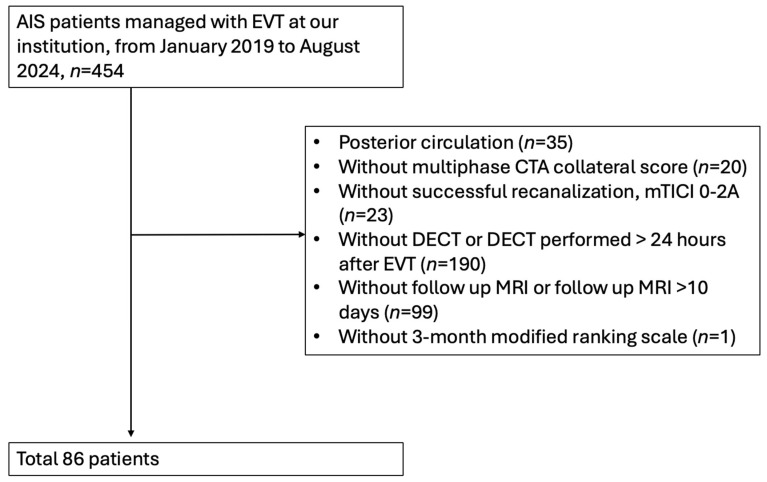
Flow-chart of inclusion and exclusion for patients with AIS.

**Figure 2 diagnostics-15-02701-f002:**
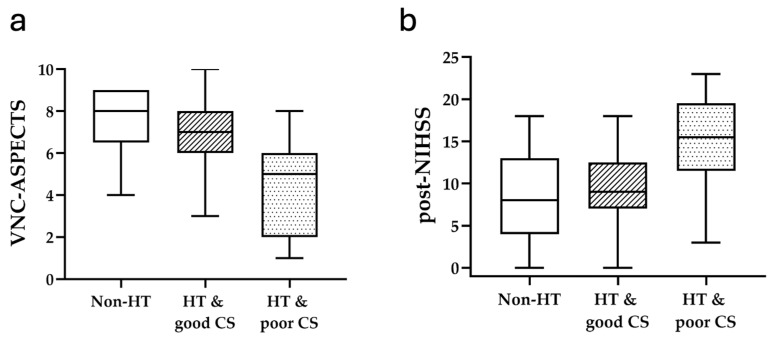
(**a**) VNC-ASPECTS: non-HT vs. HT and good CS vs. HT and poor CS (8 vs. 6 vs. 5, *p* < 0.001); (**b**) Post-24 h NIHSS: non-HT vs. HT and good CS vs. HT and poor CS (8 vs. 10 vs. 15.5, *p* = 0.002).

**Figure 3 diagnostics-15-02701-f003:**
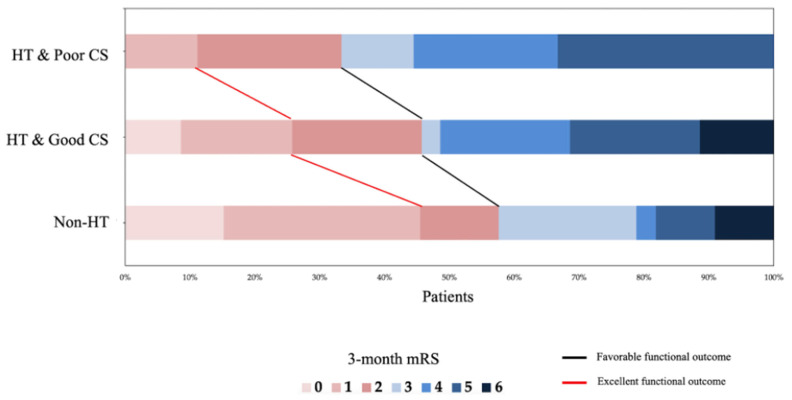
Distribution of 3-month mRS by hemorrhagic transformation (HT) and collateral status (CS) strata. Horizontal 100% stacked bars show the proportion of patients at each mRS level (0–6; color key). Groups are non-HT (*n* = 33), HT with good CS (collateral score 4–5; *n* = 35), and HT with poor CS (score 0–3; *n* = 18). The red polyline marks the cumulative proportion achieving excellent outcome (mRS ≤ 1); the black polyline marks favorable outcome (mRS ≤ 2). Favorable outcome rates were 58%, 46%, and 33% (*p* = 0.242); excellent outcome rates were 46%, 26%, and 11% for the three groups, respectively (*p* = 0.029). Abbreviations: HT, hemorrhagic transformation; CS, collateral status; mRS, modified Rankin Scale.

**Figure 4 diagnostics-15-02701-f004:**
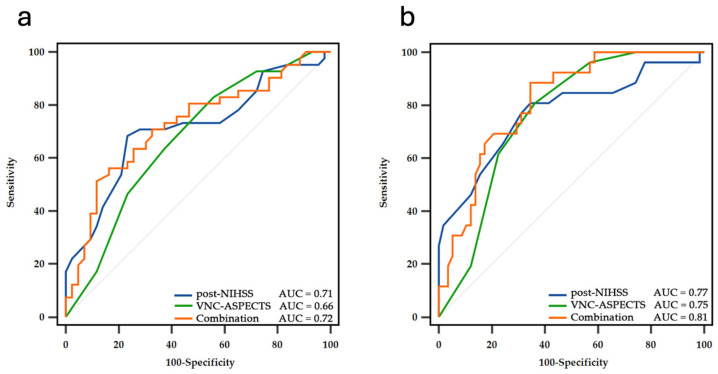
ROC curve of functional outcome. (**a**) ROC curve of favorable outcome: combined score using both post-EVT NIHSS and VNC-ASPECTS showing an area under the curve of 0.724, with 56% sensitivity and 84% specificity; (**b**) ROC curve of excellent outcome: the combined score achieved an AUC of 0.813, with 88% sensitivity and 65% specificity.

**Table 1 diagnostics-15-02701-t001:** Clinical and imaging study analyses of hemorrhagic transformation.

	Total (*n* = 86)	Hemorrhagic Transformation		*p* Value
		Non-HT (*n* = 33)	HT (*n* = 53)	
Demographic information				
Age (years), median (IQR)	70 (60–77)	67 (59–75)	70 (61–78.5)	0.263
Gender, *n* (%)				0.415
Female	36 (41.9%)	12 (36.4%)	24 (45.3%)	
Male	50 (58.1%)	21 (63.6%)	29 (54.7%)	
BMI (kg/m^2^), median (IQR)	24.6 (21.8–26.7)	24.8 (21.4–27.2)	23.4 (21.8–26.3)	0.461
Comorbidity, *n* (%)				
Hypertension	52 (60.5%)	18 (54.5%)	34 (64.2%)	0.376
Diabetes Mellitus	33 (38.4%)	15 (45.5%)	18 (34%)	0.287
Dyslipidemia	63 (73.3%)	24 (72.7%)	39 (73.6%)	0.930
Atrial fibrillation	44 (51.2%)	18 (54.5%)	26 (49.1%)	0.620
Previous tPA	38 (44.2%)	20 (60.6%)	18 (34%)	0.016 *
Laboratory data, median (IQR)				
Hb (g/dL)	13.9 (12.2–15)	13.7 (11.6–15)	13.9 (12.5–15.2)	0.531
WBC (cells/μL)	8165 (6335–11,190)	8520 (6010–11,525)	8150 (6490–9910)	0.739
Platelet (×10^3^/μL)	205.5 (162.3–248.8)	189 (168.5–249.5)	210 (158.5–250.5)	0.736
Cholesterol (mg/dL)	152 (133.5–185)	148 (127–170)	164 (139.3–209.8)	0.013 *
LDL (mg/dL)	92.5 (73–123.8)	88 (67.5–103.5)	102 (77–139)	0.012 *
Reperfusion time, median (IQR)				
Onset to reperfusion (hours)	6.3 (4.5–8.8)	5.6 (4.4–7.2)	6.5 (4.5–9.7)	0.062
Door to reperfusion (hours)	2.7 (2.2–3.4)	2.5 (2.2–3.4)	2.8 (2.1–3.4)	0.644
Puncture to reperfusion (hours)	0.8 (0.5–1.2)	0.8 (0.5–1.2)	0.8 (0.5–1.2)	0.989
NIHSS, median (IQR)				
ER NIHSS	16.5 (13–20)	16 (13–18.5)	17 (12.5–21)	0.321
Post-24 h NIHSS	10 (6–16)	8 (4–13)	13 (7–17.5)	0.005 **
NIHSS improved	5 (2–10)	8 (5–12)	4 (1.5–10)	0.033 *
Clinical Outcomes				
Neurosurgical intervention, *n* (%)	5 (5.8%)	2 (6.1%)	3 (5.7%)	1.000
In hospital length, median (IQR)	14.5 (9–19)	12 (8–15)	16 (11–20.5)	0.002 **
Favorable mRS at 3-month	41 (47.7%)	19 (57.6%)	22 (41.5%)	0.147
Excellent mRS at 3-month	26 (30.2%)	15 (45.5%)	11 (20.8%)	0.015 *
3-month mortality	7 (8.1%)	3 (9.1%)	4 (7.5%)	1.000
Preprocedural CT, median (IQR)				
CBF < 30% of CTP (mL)	16 (0–43)	10 (0–40.5)	19.5 (0–43)	0.860
Tmax > 6 s of CTP (mL)	98 (68–131)	100 (74–138)	93.5 (62.3–128.8)	0.443
Tmax > 10 s of CTP (mL)	55 (17–77)	51 (27–83.5)	56 (15.8–75.5)	0.496
Hypoperfusion index ratio	0.5 (0.3–0.7)	0.5 (0.3–0.7)	0.5 (0.3–0.7)	0.937
Mismatch ratio	3.2 (2.3–7.4)	2.7 (2.4–13.5)	3.3 (2.1–5.8)	0.516
Collateral score	4 (3–4)	4 (3–4)	4 (3–4)	0.382
NCCT-ASPECTS	9 (8–9)	9 (8–9)	8 (8–9)	0.005 **
Procedural, *n* (%)				
Occlusion side				0.797
Left	51 (59.3%)	19 (57.6%)	32 (60.4%)	
Right	35 (40.7%)	14 (42.4%)	21 (39.6%)	
Occlusion location				0.859
MCA	62 (72.1%)	24 (72.7%)	38 (71.7%)	
ICA	9 (10.5%)	4 (12.1%)	5 (9.4%)	
Combined (MCA/ICA/ACA)	15 (17.4%)	5 (15.2%)	10 (18.9%)	
mTICI				0.769
2B	22 (25.6%)	8 (24.2%)	14 (26.4%)	
2C	13 (15.1%)	4 (12.1%)	9 (17%)	
3	51 (59.3%)	21 (63.6%)	30 (56.6%)	
PTA or Stent				0.338
Non	68 (79.1%)	28 (84.8%)	40 (75.5%)	
PTA	11 (12.8%)	2 (6.1%)	9 (17%)	
PTA with stent	7 (8.1%)	3 (9.1%)	4 (7.5%)	
Post-procedural DECT, median (IQR)				
sNCCT-ASPECTS	8 (6–9)	9 (7.5–9)	7 (6–8)	<0.001 **
VNC-ASPECTS	6.5 (5–8)	8 (6.5–9)	6 (5–7)	<0.001 **
IOM-ASPECTS	6 (6–8)	5 (3–6.5)	7 (6–8)	0.052
Post-procedural MRI				
Ischemic volume (mL) by MRI, median (IQR)	16.6 (5.1–41.6)	15.4 (5.4–44.3)	16.7 (4.8–42.3)	0.773
DWI-ASPECT, median (IQR)	6 (4–7)	6 (5–8)	5 (4–6)	0.002 **
Image outcomes				
NCCTDWI, median (IQR)	3 (1–4)	3 (1–4)	3 (2–4)	0.064
VNCDWI, median (IQR)	0 (0–2)	1 (0–2)	0 (0–1)	0.348
Ischemic volume trend (mL), median (IQR)	3.3 (−15.4–19.9)	2.6 (−10.7–24)	3.6 (−19.6–22)	1.000

BMI, body mass index; Hb, hemoglobin; WBC, white blood cell count; LDL, low-density lipoprotein; ER, emergency room; NIHSS, National Institutes of Health Stroke Scale; mRS, modified Rankin Scale; NCCT, non-contrast CT; sNCCT, simulated 120-kV NCCT; VNC, virtual non-contrast; IOM, iodine overlay map; DWI, diffusion-weighted imaging; CTP, CT perfusion; CBF, cerebral blood flow; ASPECTS, Alberta Stroke Program Early CT Score; mTICI, modified Thrombolysis in Cerebral Infarction; PTA, percutaneous transluminal angioplasty; IQR, interquartile range. Statistical tests: Mann–Whitney U; Chi-square or Fisher–Freeman–Halton, as appropriate. Significance: * *p* < 0.05. ** *p* < 0.01.

**Table 2 diagnostics-15-02701-t002:** Hemorrhagic transformation and collateral score clinical and imaging analysis.

	Total (*n* = 86)	Hemorrhagic Transformation and Collateral Score			*p* Value
		Non-HT (*n* = 33)	HT and Good CS (*n* = 35)	HT and Poor CS (*n* = 18)	
Demographic information					
Age (years), median (IQR)	70 (60–77)	67 (59–75)	70 (61–78)	71 (60.5–79)	0.506
Gender, *n* (%)					0.129
Female	36 (41.9%)	12 (36.4%)	19 (54.3%)	5 (27.8%)	
Male	50 (58.1%)	21 (63.6%)	16 (45.7%)	13 (72.2%)	
BMI (kg/m^2^), median (IQR)	24.6 (21.8–26.7)	24.8 (21.4–27.2)	22.7 (21.1–26.1)	25.4 (22.5–28.3)	0.138
Comorbidity, *n* (%)					
Hypertension	52 (60.5%)	18 (54.5%)	23 (65.7%)	11 (61.1%)	0.641
Diabetes Mellitus	33 (38.4%)	15 (45.5%)	12 (34.3%)	6 (33.3%)	0.565
Dyslipidemia	63 (73.3%)	24 (72.7%)	26 (74.3%)	13 (72.2%)	0.983
Atrial fibrillation	44 (51.2%)	18 (54.5%)	16 (45.7%)	10 (55.6%)	0.703
Previous tPA	38 (44.2%)	20 (60.6%)	10 (28.6%)	8 (44.4%)	0.029 *
Laboratory data, median (IQR)					
Hb (g/dL)	13.9 (12.2–15)	13.7 (11.6–15)	13.9 (12.1–15.5)	13.9 (13–14.9)	0.816
WBC (cells/μL)	8165 (6335–11,190)	8520 (6010–11,525)	8180 (6640–10,220)	7890 (5775–9595)	0.642
Platelet (×10^3^/μL)	205.5 (162.3–248.8)	189 (168.5–249.5)	219 (165–275)	171.5 (141.5–215)	0.088
Cholesterol (mg/dL)	152 (133.5–185)	148 (127–170)	169.5 (142–208.8)	152 (134.5–210.8)	0.040 *
LDL (mg/dL)	92.5 (73–123.8)	88 (67.5–103.5)	102 (79–141)	98.5 (77–137.5)	0.041 *
PT (s)	10.8 (10.5–11.3)	10.6 (10.5–11.2)	10.7 (10.4–11.3)	11.2 (10.8–11.9)	0.024 *
APTT (s)	26.8 (24.6–28.8)	27.2 (25–30.3)	24.9 (23.6–27.3)	27.1 (25.4–29.5)	0.012 *
Reperfusion time, median (IQR)					
Onset to reperfusion (h)	6.3 (4.5–8.8)	5.6 (4.4–7.2)	6.9 (5.7–10)	5 (4.4–8)	0.020 *
Door to reperfusion (h)	2.7 (2.2–3.4)	2.5 (2.2–3.4)	2.8 (2.3–3.5)	2.4 (2–3.4)	0.448
Puncture to reperfusion (h)	0.8 (0.5–1.2)	0.8 (0.5–1.2)	0.9 (0.5–1.3)	0.6 (0.4–1.2)	0.712
NIHSS, median (IQR)					
ER NIHSS	16.5 (13–20)	16 (13–18.5)	16 (11–20)	18 (13–24.3)	0.351
Post-24 h NIHSS	10 (6–16)	8 (4–13)	10 (7–16)	15.5 (11.5–19.5)	0.002 **
NIHSS improved	5 (2–10)	8 (5–12)	4 (2–10)	4.5 (−0.5–7.8)	0.082
Clinical Outcomes					
Neurosurgical intervention, *n* (%)	5 (5.8%)	2 (6.1%)	1 (2.9%)	2 (11.1%)	0.349
In hospital length, median (IQR)	14.5 (9–19)	12 (8–15)	15 (11–20)	18 (14–21.3)	0.005 **
Favorable mRS at 3-month	41 (47.7%)	19 (57.6%)	16 (45.7%)	6 (33.3%)	0.242
Excellent mRS at 3-month	26 (30.2%)	15 (45.5%)	9 (25.7%)	2 (11.1%)	0.029 *
3-month mortality	7 (8.1%)	3 (9.1%)	4 (11.4%)	0 (0%)	0.515
Preprocedural CT, median (IQR)					
CBF < 30% of CTP (mL)	16 (0–43)	10 (0–40.5)	7.5 (0–25.3)	43 (33–56.3)	0.004 **
Tmax > 6 s of CTP (mL)	98 (68–131)	100 (74–138)	85 (46.3–128.8)	100.5 (80–126.5)	0.523
Tmax > 10 s of CTP (mL)	55 (17–77)	51 (27–83.5)	47 (10–75.5)	63.5 (43.5–78.8)	0.335
Hypoperfusion index ratio	0.5 (0.3–0.7)	0.5 (0.3–0.7)	0.4 (0.2–0.6)	0.7 (0.5–0.7)	0.106
Mismatch ratio	3.2 (2.3–7.4)	2.7 (2.4–13.5)	5.2 (2.8–7.3)	2.3 (1.6–3.5)	0.023 *
Collateral score	4 (3–4)	4 (3–4)	4 (4–5)	3 (2–3)	<0.001 **
NCCT-ASPECTS	9 (8–9)	9 (8–9)	8 (8–9)	8 (7–9)	0.015 *
Procedural, *n* (%)					
Occlusion side					0.965
Left	51 (59.3%)	19 (57.6%)	21 (60%)	11 (61.1%)	
Right	35 (40.7%)	14 (42.4%)	14 (40%)	7 (38.9%)	
Occlusion location					0.765
MCA	62 (72.1%)	24 (72.7%)	26 (74.3%)	12 (66.7%)	
ICA	9 (10.5%)	4 (12.1%)	4 (11.4%)	1 (5.6%)	
Combined (MCA/ICA/ACA)	15 (17.4%)	5 (15.2%)	5 (14.3%)	5 (27.8%)	
mTICI					0.941
2B	22 (25.6%)	8 (24.2%)	10 (28.6%)	4 (22.2%)	
2C	13 (15.1%)	4 (12.1%)	6 (17.1%)	3 (16.7%)	
3	51 (59.3%)	21 (63.6%)	19 (54.3%)	11 (61.1%)	
PTA or Stent					0.269
Non	68 (79.1%)	28 (84.8%)	24 (68.6%)	16 (88.9%)	
PTA	11 (12.8%)	2 (6.1%)	7 (20%)	2 (11.1%)	
PTA with stent	7 (8.1%)	3 (9.1%)	4 (11.4%)	0 (0%)	
Post-procedural DECT, median (IQR)					
sNCCT-ASPECTS	8 (6–9)	9 (7.5–9)	7 (7–9)	6 (4–8)	<0.001 **
VNC-ASPECTS	6.5 (5–8)	8 (6.5–9)	6 (5–7)	5 (2–6)	<0.001 **
IOM-ASPECTS	6 (6–8)	5 (3–6.5)	7 (6–8)	6 (5.3–6.8)	0.059
Post-procedural MRI					
Ischemic volume (mL) by MRI, median (IQR)	16.6 (5.1–41.6)	15.4 (5.4–44.3)	14.4 (4.7–37.9)	27.0 (4.9–61.8)	0.589
DWI-ASPECT, median (IQR)	6 (4–7)	6 (5–8)	6 (5–6)	4 (2–5.3)	<0.001 **
Image outcomes					
NCCTDWI, median (IQR)	3 (1–4)	3 (1–4)	2 (1–4)	4 (3–5)	0.003 **
VNCDWI, median (IQR)	0 (0–2)	1 (0–2)	0 (0–1)	0 (0–1.3)	0.632
Ischemic volume trend (mL), median (IQR)	3.3 (−15.4–19.9)	2.6 (−10.7–24)	4.1 (−14.7–19.4)	0.7 (−30.8–63.2)	0.949

BMI, body mass index; Hb, hemoglobin; WBC, white blood cell count; LDL, low-density lipoprotein; PT, prothrombin time; APTT, activated partial thromboplastin time; ER, emergency room; NIHSS, National Institutes of Health Stroke Scale; mRS, modified Rankin Scale; NCCT, non-contrast CT; sNCCT, simulated 120-kV NCCT; VNC, virtual non-contrast; IOM, iodine overlay map; DWI, diffusion-weighted imaging; CTP, CT perfusion; CBF, cerebral blood flow; ASPECTS, Alberta Stroke Program Early CT Score; mTICI, modified Thrombolysis in Cerebral Infarction; PTA, percutaneous transluminal angioplasty; CS, collateral status; IQR, interquartile range. Statistical tests: Kruskal–Wallis; Chi-square or Fisher–Freeman–Halton, as appropriate. Significance: * *p* < 0.05. ** *p* < 0.01.

**Table 3 diagnostics-15-02701-t003:** Predictive factors of favorable mRS at 3 months.

	Simple Model			Multiple Model		
	OR	(95% CI)	*p* Value	OR	(95% CI)	*p* Value
Hemorrhagic transformation	0.52	(0.22–1.26)	0.149			
Hemorrhagic transformation and collateral score						
Non-HT	1.00					
HT and good CS	0.62	(0.24–1.62)	0.329			
HT and poor CS	0.37	(0.11–1.22)	0.103			
Age	0.95	(0.91–0.99)	0.007 **	0.95	(0.91–0.99)	0.011 *
ER NIHSS	0.86	(0.79–0.95)	0.002 **			
Post-24 h NIHSS	0.89	(0.82–0.96)	0.003 **	0.94	(0.86–1.02)	0.136
Onset to reperfusion (h)	1.03	(0.92–1.14)	0.639			
NCCT-ASPECTS	1.36	(0.87–2.12)	0.172			
sNCCT-ASPECTS	1.30	(1.02–1.67)	0.036 *			
VNC-ASPECTS	1.33	(1.07–1.66)	0.012 *	1.32	(0.99–1.74)	0.051
Ischemic volume (mL) by MRI	1.00	(1.00–1.00)	0.105			
DWI-ASPECT	1.23	(0.98–1.54)	0.070			
NCCTDWI	0.85	(0.67–1.09)	0.201			
VNCDWI	1.28	(0.92–1.76)	0.140			
Ischemic volume trend	0.99	(0.98–1.00)	0.209			

Logistic regression. * *p* < 0.05, ** *p* < 0.01.

**Table 4 diagnostics-15-02701-t004:** Predictive factors of excellent mRS at 3 months.

	Simple Model			Multiple Model		
	OR	(95% CI)	*p* Value	OR	(95% CI)	*p* Value
Hemorrhagic transformation	0.31	(0.12–0.82)	0.017 *			
Hemorrhagic transformation and collateral score						
Non-HT	1.00			1.00		
HT and good CS	0.42	(0.15–1.15)	0.092	0.72	(0.21–2.46)	0.595
HT and poor CS	0.15	(0.03–0.76)	0.022 *	0.75	(0.11–5.31)	0.770
Age	0.96	(0.92–0.998)	0.040 *			
ER NIHSS	0.87	(0.79–0.96)	0.006 **			
Post-24 h NIHSS	0.84	(0.76–0.93)	0.001 **	0.89	(0.80–0.99)	0.027 *
Onset to reperfusion (h)	0.97	(0.86–1.10)	0.641			
NCCT-ASPECTS	1.74	(1.02–2.97)	0.043 *			
sNCCT-ASPECTS	1.88	(1.26–2.83)	0.002 **			
VNC-ASPECTS	1.79	(1.28–2.52)	0.001 **	1.56	(0.89–2.71)	0.118
Ischemic volume (mL) by MRI	1.00	(1.00–1.00)	0.146			
DWI-ASPECT	1.59	(1.20–2.11)	0.001 **	1.04	(0.66–1.64)	0.865
NCCTDWI	0.67	(0.49–0.91)	0.010 *			
VNCDWI	1.17	(0.83–1.64)	0.367			
Ischemic volume trend	1.00	(0.98–1.01)	0.625			

Logistic regression. * *p* < 0.05, ** *p* < 0.01.

## Data Availability

Data are available on request due to restrictions (e.g., privacy, legal, or ethical reasons). The data presented in this study are available on request from the corresponding author due to IRB restriction.

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
