# Peer review of "Impact of Preprocedural Collateral Status on Hemorrhagic Transformation and Outcomes After Endovascular Thrombectomy in Acute Ischemic Stroke [Author-notes fn1-diagnostics-15-02701]"

_diagnostics, 2025, doi:10.3390/diagnostics15212701_

Round 1

Reviewer 1 Report

Comments and Suggestions for Authors

1.On the 10th day, regarding the assessment of  hemorrhagic transformation,whether the magnetic resonance susceptibility sequence was applied, it would be best to clearly elaborate in the text on the sequence and method used for the assessment.

2.For patients who underwent mechanical thrombectomy for anterior circulation cerebral infarction, the relationship between the infarction site and the occurrence of hemorrhagic transformation was not mentioned in the article. Comparisons need to be made among different sites.

Reviewer 2 Report

Comments and Suggestions for Authors

The manuscript entitled “Impact of Pre-procedural Collateral Status on Hemorrhagic Transformation and Outcomes After Endovascular Thrombectomy in Acute Ischemic Stroke” analyzes the importance of collateral status and imaging biomarkers in predicting outcomes after EVT for acute ischemic stroke.

These results align with prior literature showing that good collaterals predict less infarct growth, a higher chance of favorable outcome, and reduced HT after reperfusion. Therefore, this study does not bring any scientifically significant information

Comments:

  • The authors did not use the standardised template for writing the manuscript
  • Tables are not written in format Diagnostics
  • Claim reference missing - line 52
  • Why was cone beam CT used initially and then dual CT after 24
  • Not all abbreviations in the text (line 25 and line 30) in the abstract are explained.
  • Table 1 is missing some physical units, e.g., age and laboratory data.
  • Figure 2: the legend is not clearly visible
  • Table 1 does not have clearly written-out abbreviations at the bottom, nor the type of statistical analysis, nor written significance.
  • Table 2 does not have some of the physical units written for all rows.
  • Tables 1 and 2 unnecessarily repeat the same demographic data and have not been reviewed. They should be reformulated to be clearer and without unnecessary repetition of data.
  • The references are not written according to the instructions for Diagnostics
  • The authors explained the limitations of their study, but the relatively small number of patients in the study may have limited the statistical power to detect certain associations, such as the relationship between poor CS and HT.

Reviewer 3 Report

Comments and Suggestions for Authors

A retrospective analysis of the incidence of hemorrhagic transformation (HT) in 86 patients with ischemic stroke treated endovascularly.
The authors assessed the impact of collateral circulation on the incidence of transformation.
The detailed analysis will take into account numerous parameters related to pre- and post-procedure flow assessment, pharmacological treatment, and laboratory results.
The authors concluded that in patients with successfully recanalized AIS, HT with poor CS was strongly associated with poorer functional outcomes and imaging scores, suggesting more severe ischemic injury and neurological deterioration. Post-EVT NIHSS and VNC-ASPECTS have emerged as key predictors of long-term prognosis, highlighting their potential as imaging biomarkers for risk stratification in clinical practice.
The paper is interesting, with extensive discussion and numerous references.
Notes:
1/ It is difficult to assess the impact, but in the HT group, a total of 13 patients underwent PTA or PTA+stent procedures (including 11 in the HT&good CS group and 2 in the HT&poor CS group). This, compared to the 5 procedures (5/33) in the non-HT group, does not constitute a significant difference. However, due to the relatively large differences in group size and small sample sizes, it may affect long-term results and indicate different degrees of progression of lesions in the treated arteries – please comment;
2/ More importantly, there is a significant difference in the amount of necrosis in the data between the groups:
a/ non-HT - 10 ml
b/ HT&good CS - 7.5 ml
c/ HT&poor CS - 43 ml (Table 2).
This may indicate that HT does not affect long-term results, but rather the initial amount of necrosis, exacerbated by subsequent HT. Hence the question – I wonder what the HT&goodCS group would look like after excluding patients who required PTA or PTA+stent treatment (reducing the rate of transformation??)?
3/ The study assesses the impact of collateral circulation on the likelihood of HT, but the group that constitutes the most important part of the study is the smallest and has the worst baseline parameters – therefore, the likelihood of the worst early and long-term outcomes is obvious and presented in available studies.

Round 2

Reviewer 2 Report

Comments and Suggestions for Authors

The manuscript entitled "Impact of Pre-procedural Collateral Status on Hemorrhagic Transformation and Outcomes After Endovascular Thrombectomy in Acute Ischemic Stroke" was thoroughly revised in response to reviewer feedback. The authors clearly identified all corrections by highlighting the specific changes within the manuscript.   They have included relevant details in the text and explained their intention to contribute the practice-oriented evidence within a specific cohort of patients with successful recanalization. Due to the improved writing, the manuscript is now suitable for publication and should be considered for acceptance.